# Breast Cancer Screening and Perceptions of Harm among Young Adults in Japan: Results of a Cross-Sectional Online Survey

Zhengai Cui [1],*, Hiromi Kawasaki [1], Miwako Tsunematsu [1], Yingai Cui [2], Md Moshiur Rahman [1], Satoko Yamasaki [1], Yuan Li [3] and Masayuki Kakehashi [1]

1 Graduate School of Biomedical and Health Sciences, Hiroshima University, Hiroshima 734-8551, Japan
2 School of Nursing, Guangdong Medical University, Dongguan 523808, China
3 Financial Department, Guangdong Medical University, Dongguan 523808, China
* Correspondence: cuiclara0502@163.com

**Abstract:** Breast cancer is the most commonly diagnosed female cancer and the leading cause of cancer death. Early detection and treatment are important to reduce the number of deaths. Japan recommends mammography every two years for women over 40 years of age. However, in recent years, an increasing number of younger women have been undergoing breast cancer screening (BCS). To reduce the harms of BCS among young adults, our study extracted data from an online survey conducted in 2018 and applied $\chi^2$ tests and logistic analysis to identify the influencing factors regarding interest in undergoing BCS. The results of our analysis support the need for a reduction in the BCS rate through awareness regarding the harms of health screening among young people. In particular, for those who receive BCS through occupational screening, we believe that improving education on breast awareness, the accuracy of occupational screening, and breast self-examination methods could reduce the harms from BCS in younger age groups.

**Keywords:** prevention and screening in breast cancers; breast cancer screening; harms; young adults





## 1. Introduction

Breast cancer (BC) is the most common female-specific cancer, with 2,261,419 cases and 684,996 deaths worldwide in 2020 [1]. In Japan, 97,142 women were diagnosed with BC in 2019, and 14,650 died in 2020 [2]. To reduce the mortality rate of BC, early detection through breast cancer screening (BCS) and appropriate treatment are important.

Prior studies have shown that mammography (MG) alone or in combination with visual inspection is effective in reducing BC mortality through the early detection of BC [3,4]. However, harms such as radiation exposure, false positives, and overdiagnosis characterize using MG for BCS.

Radiation exposure can accumulate throughout a woman's lifetime and increase the risk of BC [5]. Additionally, high breast tissue density in young women is a likely cause of false positives [6]. The risk of a false-positive MG is about 20% for women in Europe who undergo biennial screening, and for those between 50 and 69 years of age this is about 20% over that period, with the risk of undergoing a biopsy as a result of a false-positive screening standing at 3% [7]. In the United States, the 10-year false-positive rate is 30%, with 50% of women experiencing a false positive once [8,9]. On the other hand, in Japan, the rate of false positives is reported to be around 10% per examination [10].

Such overdiagnosis can lead to an increase in unnecessary detailed examinations and treatment through detecting cancers that are not inherently prognostic for life due to the sensitivity of the test, the age at which the test is started and completed, and the effect of the interval between examinations [11]. Recent studies have shown that the overdiagnosis rate for breast cancer varies widely, ranging from 0% to 54% [12]. Studies based on statistical models consistently estimate the overdiagnosis rates to be below 5% [13,14]. In contrast,

observational studies have published higher estimates ranging from 22% to 54%, depending on the denominator used [12,14,15].

The BCS guideline recommends reducing the harms from MG by adjusting the age at which BCS is conducted. Accordingly, the U.S. Preventive Services Task Force (USPSTF) suggests that the age for starting BCS should begin at 50 years instead of 40 years. In Europe, in 2003, the European Council recommended that the age for BCS begin at 50 years [16]. On the other hand, in Japan, the Ministry of Health, Labour and Welfare (MHLW) recommends BCS (medical interview and MG) once every two years for women over 40 years old [17,18]. However, the effectiveness of BCS in young people is limited and does not justify the exposure to radiation, discomfort, and additional costs [4]. A number of experts have also recognized BCS as resulting in more harm than benefits, and they recommend that no examination be undertaken [19]. Since the nuclear accident at Fukushima, Japanese people have become emphatically aware of the harms of radiation [20,21].

In Japan, women are screened for BCS through both population- and workplace-based screening. Population-based screening is a public preventive measure that implements methods with established effectiveness in reducing mortality among those screened, and the benefits of screening outweigh and minimize the harms. On the other hand, workplace-based screening is provided by insurers for the purpose of welfare benefits, resulting in comprehensive medical checkups conducted at medical institutions; however, the age of the persons to be examined, examination methods, and the minimization of harms cannot be ensured [22]. Moreover, approximately 3800 BC patients younger than 40 years of age are affected annually, which is 4% of the total number of BC patients in all age groups [23].

The purpose of BCS is to provide benefits, such as a reduction in BC mortality. However, in recent years, there has been a need to consider the benefits and harms of BCS and to evaluate its overall effectiveness. Compared with other age groups, the risk of harm from BCS among young adults is considerable [19]. A risk stratification (young, middle-aged, and older adults) of BCS may lead to a reduction in harm, an increase in screening quality, and superior cost-effectiveness [24]. To the best of our knowledge, the factors influencing young-age BCS, including its harms, have not been elucidated.

Attitudes and beliefs about BCS are significant factors when deciding whether or not to undergo health screening. The general public's attitude toward BCS is that there are associated harms; however, younger adults are less perceptive about the risks of BCS. On the other hand, much of the literature on health beliefs and preventive health behaviors towards BCS is based on theoretical constructs, such as the health belief model (HBM) and rational behavior, which consider health-related beliefs and perceived barriers to health-promoting behaviors. HBM is also the most used method to explain early diagnosis behaviors for BCS [25]. We have already acknowledged the involvement of the HBM with respect to participation in uterine cancer screening [26]. Its involvement in BCS is not yet clear.

Therefore, this study uses the HBM as a theoretical framework with the aims of (i) identifying the factors that influence BCS and (ii) identifying the factors that influence the perception of harm about receiving BCS in young adults. We also fill a gap in the literature regarding the uptake of BCS among young adults according to the interest in BCS harms.

## 2. Materials and Methods

### 2.1. Analysis Target

We analyzed data from an online survey conducted in 2018 (Approval No. E-1081-1). Our selection criteria for the analyzed data were: (i) no history of either breast or cervical cancer; (ii) patients in the 20–30 age group.

The title of the online survey was "2018 Attitude Survey on Factors Influencing Women's Cancer Screening Behavior". In the survey, an online research firm created a survey instrument for users and sent an email requesting the survey. The survey participants accessed the URL provided in the email and completed the survey. The criteria for the

survey targets were (1) women living in Japan, (2) women between the ages of 20 and 69, and (3) women who agreed to participate in the survey.

The survey used "registered monitors" owned by an online research company (Hiroshima, Japan). Samples were drawn according to the percentage of the population (see 2015 Census) in each age category (20 s, 30 s, 40 s, 50 s, and 60 s). The survey was stopped when the target sample for each category was reached. A sample of approximately 3000 persons was deemed adequate.

### 2.2. Data Collection

The personal characteristics, BCS participation status, reasons for non-participating in BCS, knowledge of the harms of cancer screening, and HBM used in this study were all derived from the "2018 Attitude Survey on Factors Influencing Women's Cancer Screening Behavior".

We constructed personal characteristics based on previous studies [27–30]: age (20–24 years, 25–29 years, 30–34 years, and 35–39 years); marital status (married, single); employment status (self-employed, regular employment, part-time job, student, housewife, or unemployed); educational background (primary and secondary school, high school, junior colleges and vocational schools, university, graduate school, etc.); household income (no income, less than USD 7000, USD 7000–22,000, USD 22,000–37,000, USD 37,000–51,000, USD 51,000–73,000, USD 73,000–110,000, and more than USD 110,000); medical insurance (association, national, or union health insurance, mutual aid association, national health insurance association, unknown, etc.) (Supplementary Material S1); medical insurance (dependent, personal, or family); medical consultation (yes, no); regular health checkups (yes, no); and private medical insurance (yes, no) (Supplementary Table S1).

The BCS participation status [31] was evaluated by a history of screening participation in the past 2 years; types of participation (population- or workplace-based, individual complete physical examination/hospital visit, etc.); and the reasons for BCS (information from municipality or workplace, recommendations from family doctor, personal healthcare, feeling the need to see a doctor, because a family member or acquaintance has cancer, etc.).

The reasons for not participating in BCS [31] options were busy, healthy, anxiety about the results, lack of awareness about cancer screening, lack of opportunity to have a cancer screening, forgetting to take the test, self-perception of not being old enough to have a checkup, participation in cervical cancer screening, etc.).

Knowledge of the harms of cancer screening was evaluated by the questions, 'Do you know about the harms of cancer screening?' (1: I do not know at all; 2: I do not know so much; 3: I cannot say either; 4: I know; 5: I know very much) and 'Do you want to know about the harms of cancer screening?' (1: I do not want to know at all; 2: I do not want to know so much; 3: I cannot say either; 4: I want to know; 5: I want to know very much).

HBM: The questionnaire survey was composed of 7 components based on the revised HBM by Hata [32] (Table 1). The HBM items survey consisted of 27 question components; each item was answered using a 5-point Likert scale (1: strongly disagree; 2: disagree; 3: neither agree nor disagree; 4: agree; 5 strongly agree). Cronbach's alpha coefficient, a measure of internal consistency among the questions comprising each scale, was 0.816 for "susceptibility to cancer", 0.820 for "seriousness of cancer", 0.885 for "benefits of cancer screening", 0.725 for "burden to participation before cancer screening", 0.704 for "burden to participation at the time of cancer screening", 0.821 for "importance of cancer screening", and 0.743 for "cues to participation in screening".

**Table 1.** Component items based on HBM (27 items).

| (1) Susceptibility to cancer | I may develop female cancer in the future. I may develop female cancer within a few years. I am more likely to develop female cancer than other women. |
|---|---|
| (2) Seriousness of cancer | I am afraid of developing female cancer. If I develop female cancer, my life will be changed. If I develop female cancer, my activities of daily living will be limited. If I develop female cancer, my family will be affected negatively. I'm scared that I'll find a woman's cancer by having a cancer screening. |
| (3) Benefits of cancer screening | Participation in female cancer screening can lead to reduction in mortality from female cancer. Participation in female cancer screening can lead to early detection of female cancer. Having a women's cancer screening can give me peace of mind about my health. Participation in female cancer screening can lead to better management of my health. |
| (4) Burden to participation before cancer screening | I do not have time to participate in female cancer screening. Participation in female cancer screening is costly. I don't know where to go for a female cancer screening. I forget to regularly participate in female cancer screening. |
| (5) Burden to participation at the time of cancer screening | I am embarrassed about participating in female cancer screening because it includes examination of a delicate area. Female cancer screening causes discomfort, even pain. I do not want to participate in female cancer screening that is performed by male doctors/staff members. |
| (6) Importance of cancer screening | There are other things in my life that are more important than getting screened for women's cancer. I do not need to participate in female cancer screening because I can visit a medical institution whenever there is any concern. I do not need to participate in female cancer screening because I have no particular subjective symptoms. Participation in female cancer screening is less important than other health issues. |
| (7) Cues to participation in screening | A close friend or acquaintance recommends that I have a cancer screening for women. My closest family members recommend that I participate in female cancer screening. The doctors at the hospital which I regularly visit recommend that I participate in female cancer screening. My close friends/acquaintances recommend that I participate in female cancer screening. |

*2.3. Statistical Analysis*

We checked the missing data status of the analyzed data (0.00%). To examine the association between the BCS and personal characteristics and perceptions of harms, we used the $\chi^2$ test. Next, we performed basic tabulations to examine the factors that contribute to participation or non-participation in BCS. Finally, a logistic analysis was conducted to identify factors influencing BCS and adverse perception. The flow of the logistic analysis in our study was as follows: (i) We conducted a correlation analysis based on the results of the $\chi^2$ test ($p < 0.05$) and the perception of harms. (ii) We conducted a logistic regression analysis with the BCS (1: participation in BCS, 0: non-participation in BCS) as the dependent variable, and the variable with $|r| < 0.5$ obtained from the correlation analysis and the seven factors of HBM as explanatory variables. (iii) We conducted a logistic analysis with the presence or absence of harms interest as the dependent variable and the variable

obtained from (ii) ($p < 0.05$) as the explanatory variable. The analysis results are presented as (adjusted) odds ratios (ORs) and 95% confidence intervals (CIs).

We performed the statistical analysis using R version 4.0.2 software, with a significance level of 5%.

## 3. Results

A total of 3249 people responded to the questionnaire. In accordance with the exclusion criteria, (i) 208 persons with a medical history of breast or cervical cancer were excluded, in addition to (ii) 1970 persons over 40 years of age, leaving a total of 1071 subjects in the analysis. Of those analyzed, 255 (23.8%) were screened, with a mean age of 32.82 years (SD $\pm$ 13.30), and 816 (71.2%) were unscreened, with a mean age of 30.48 years (SD $\pm$ 13.70) (Figure 1).

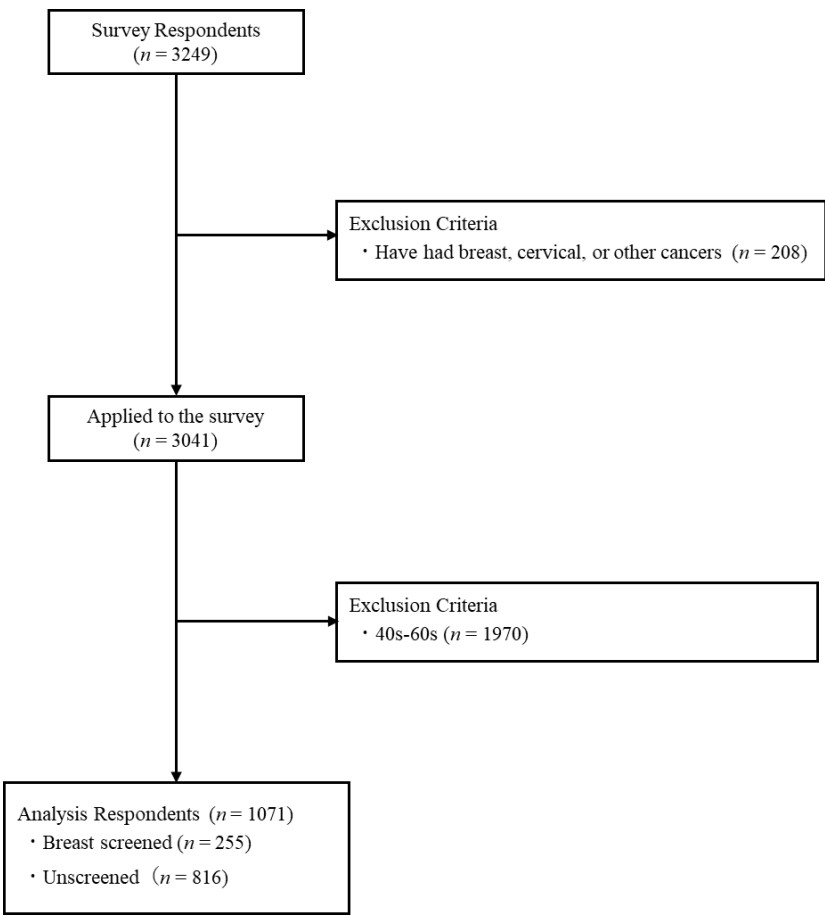

**Figure 1.** Selection criteria for analytic subjects' information.

### 3.1. Factors Affecting Participation in BCS

We analyzed the association between the BCS and individual characteristics and harms: age, marital status, work status, educational background, household income, medical insurance, medical insurance (dependent), medical consultation, having regular health checkups, and private medical insurance. We observed significant differences.

Observing the number of respondents by item showed that the numbers for an awareness (I know, I know very much) of harms were 12 (24.5%) and 37 (75.5%) for those who did or did not receive BCS, respectively. On the other hand, 194 (24.2%) and 607 (75.8%) of the examined and unexamined respondents were concerned about the harms (I want to know, I want to know very much) of BCS, respectively.

Next, among the personal characteristics, 11 (10.2%) of the examinees were in the 20–24 age group, 12 (13.8%) were in the 25–29 age group, 93 (30.6%) were in the 30–34 age

group, and 104 (32.6%) were in the 35–39 age group. Employed subjects comprised the highest percentage of respondents, with 138 (28.9%) in regular employment, 38 (20.0%) in part-time employment, and 63 (23.5%) replying that they were a housewife. Additionally, 131 (30.5%) were college graduates. A total of 54 (16.9%) responded that they had a household income of USD 22,000–USD 37,000, while 73 (25.7%) had an income of USD 37,000–USD 51,000, and 65 (34.4%) had an income of USD 51,000–USD 73,000. Of those with medical insurance, 90 (22.0%) were covered by association health insurance, 64 (37.4%) by union health insurance, and 49 (17.4%) by national health insurance. In addition, 110 (44.7%) regularly underwent health checkups (Table 2).

**Table 2.** Associations between BCS behaviors and personal characteristics/harms.

| Characteristic | Total (*n* = 1071) | Screened (*n* = 255) | Unscreened (*n* = 816) | *p*-Value * |
|---|---|---|---|---|
| Do you know about the harms of cancer screening? | | | | 0.481 |
| I don't know at all | 458 (42.8%) | 97 (21.2%) | 361 (78.8%) | |
| I don't know so much | 453 (42.3%) | 115 (25.4%) | 338 (74.6%) | |
| I can't say either | 111 (10.4%) | 31 (27.9%) | 80 (72.1%) | |
| I know | 44 (4.1%) | 11 (25.0%) | 33 (75.0%) | |
| I know very much | 5 (0.5%) | 1 (20.0%) | 4 (80.0%) | |
| Do you want to know about the harms of cancer screening? | | | | 0.779 |
| I don't want to know at all | 15 (1.4%) | 5 (33.3%) | 10 (66.7%) | |
| I don't want to know so much | 24 (2.2%) | 4 (16.7%) | 20 (83.3%) | |
| I can't say either | 231 (21.6%) | 52 (22.5%) | 179 (77.5%) | |
| I want to know | 587 (54.8%) | 141 (24.0%) | 446 (76.0%) | |
| I want to know very much | 214 (20.0%) | 53 (24.8%) | 161 (75.2%) | |
| Age | | | | <0.001 |
| 20–24 | 108 (10.1%) | 11 (10.2%) | 97 (89.8%) | |
| 25–29 | 340 (31.8%) | 47 (13.8%) | 293 (86.2%) | |
| 30–34 | 304 (28.4%) | 93 (30.6%) | 211 (69.4%) | |
| 35–39 | 319 (29.8%) | 104 (32.6%) | 215 (67.4%) | |
| Marital status | | | | 0.004 |
| Married | 630 (58.8%) | 170 (27.0%) | 460 (73.0%) | |
| Single | 441 (41.2%) | 85 (19.3%) | 356 (80.7%) | |
| Work status | | | | <0.001 |
| Self-employed | 35 (3.3%) | 9 (25.7%) | 26 (74.3%) | |
| Regular employment | 478 (44.6%) | 138 (28.9%) | 340 (71.1%) | |
| Parttime job | 190 (17.7%) | 38 (20.0%) | 152 (80.0%) | |
| Student | 49 (4.6%) | 3 (6.1%) | 46 (93.9%) | |
| Housewife | 268 (25.0%) | 63 (23.5%) | 205 (76.5%) | |
| Unemployed | 51 (4.8%) | 4 (7.8%) | 47 (92.2%) | |
| Educational background | | | | 0.001 |
| Primary and secondary school | 27 (2.5%) | 5 (18.5%) | 22 (81.5%) | |
| High School | 236 (22.0%) | 40 (17.0%) | 196 (83.1%) | |
| Junior colleges and vocational schools | 213 (19.9%) | 39 (18.3%) | 174 (81.7%) | |
| University | 430 (40.2%) | 131 (30.5%) | 299 (69.5%) | |
| Graduate School | 33 (3.1%) | 9 (27.3%) | 24 (72.7%) | |
| Others | 132 (12.3%) | 31 (23.5%) | 101 (76.5%) | |
| Household income ** | | | | <0.001 |
| No income | 18 (1.7%) | 3 (16.7%) | 15 (83.3%) | |
| Less than USD 7000 | 27 (2.5%) | 4 (14.8%) | 23 (85.2%) | |
| USD 7000–USD 22,000 | 115 (10.7%) | 18 (15.7%) | 97 (84.4%) | |
| USD 22,000–USD 37,000 | 320 (29.9%) | 54 (16.9%) | 266 (83.1%) | |
| USD 37,000–USD 51,000 | 284 (26.5%) | 73 (25.7%) | 211 (74.3%) | |
| USD 51,000–USD 73,000 | 189 (17.7%) | 65 (34.4%) | 124 (65.6%) | |
| USD 73,000–USD 110,000 | 91 (8.5%) | 30 (33.0%) | 61 (67.0%) | |
| More than USD 110,000 | 27 (2.5%) | 8 (29.6%) | 19 (70.4%) | |

**Table 2.** *Cont.*

| Characteristic | Total (n = 1071) | Screened (n = 255) | Unscreened (n = 816) | *p*-Value * |
|---|---|---|---|---|
| Medical insurance *** | | | | <0.001 |
| Association health insurance | 409 (38.2%) | 90 (22.0%) | 319 (78.0%) | |
| Union health insurance | 171 (16.0%) | 64 (37.4%) | 107 (62.6%) | |
| Mutual aid association | 104 (9.7%) | 29 (27.9%) | 75 (72.1%) | |
| National health insurance | 282 (26.3%) | 49 (17.4%) | 233 (82.6%) | |
| National health insurance association | 63 (5.9%) | 13 (20.6%) | 50 (79.4%) | |
| Others | 16 (1.5%) | 4 (25.0%) | 12 (75.0%) | |
| Unknown | 26 (2.4%) | 6 (23.1%) | 20 (76.9%) | |
| Medical insurance (dependent) | | | | 0.003 |
| Myself | 638 (59.6%) | 173 (27.1%) | 465 (72.9%) | |
| Family | 433 (40.4%) | 82 (18.9%) | 351 (81.1%) | |
| Medical consultation | | | | 0.002 |
| Yes | 286 (26.7%) | 88 (30.8%) | 198 (69.2%) | |
| No | 785 (73.3%) | 167 (21.3%) | 618 (78.7%) | |
| Have regular health checkups | | | | <0.001 |
| Yes | 246 (23.0%) | 110 (44.7%) | 136 (55.3%) | |
| No | 825 (77.0%) | 145 (17.6%) | 680 (82.4%) | |
| Private medical insurance | | | | 0.001 |
| Yes | 557 (52.0%) | 157 (28.2%) | 400 (71.8%) | |
| No | 514 (48.0%) | 98 (19.1%) | 416 (80.9%) | |

* $\chi^2$ test. ** Household income: calculated according to the exchange rate on 26 July 2022 (JPY 1 = USD 0.0073). *** Employee insurance mainly includes: "Association health insurance (for employees of small and medium-sized companies and their dependents)", "Union health insurance (for employees of large companies and their dependents)", "Mutual aid association (for public employees and their dependents)", and "National health insurance association (for doctors, construction workers and their dependents)". Regional insurance includes "National health insurance (for people who are not covered by employee insurance, such as the self-employed and unemployed)".

### 3.2. Status of Participation in BCS

Table 3 shows the participation in BCS by each personal characteristic: 54 (39.1%) were in regular employment, mainly in the 30–34 age group; 56 (40.6%) had association health insurance; and 62 (44.9%) underwent BCS using "My own healthcare".

The ages of those with a part-time job were mainly concentrated in the 35–39 age group, namely 23 years of age (60.5%), while 17 (44.7%) had association health insurance, and 15 (39.5%) underwent BCS using "My own healthcare".

Housewives comprised 29 (46.0%) subjects in the 35–39 age group, while 23 (36.5%) had union health insurance and 29 (46.0%) underwent BCS using "My own healthcare" or responded as being part-time.

### 3.3. Status of Non-Participation in BCS

Table 4 shows the frequency of BCS non-participation by attribute. By age, 97 (10.6%) were in the 20–24 age group, 43 (44.3%) were students, and 41 (42.3%) had national health insurance. A total of 32 (33.0%) did not undergo BCS, responding that: "I do not think I am old enough to have a checkup". Furthermore, 23 (23.7%) responded: "Because I never had a chance to have a cancer screening".

**Table 3.** Status of participation in BCS.

| Characteristic | Total (n = 255) | Work Status | | | | | |
|---|---|---|---|---|---|---|---|
| | | Self-Employed (n = 9) | Regular Employment (n = 138) | Part-Time Job (n = 38) | Student (n = 3) | Housewife (n = 63) | Unemployed (n = 4) |
| Age | | | | | | | |
| 20–24 | 9 (3.5%) | 0 (0.0%) | 7 (5.1%) | 1 (2.6%) | 3 (100.0%) | 0 (0.0%) | 0 (0.0%) |
| 25–29 | 138 (54.1%) | 1 (11.1%) | 29 (21.0%) | 5 (13.2%) | 0 (0.0%) | 10 (15.9%) | 2 (50.0%) |
| 30–34 | 38 (14.9%) | 6 (66.7%) | 54 (39.1%) | 9 (23.7%) | 0 (0.0%) | 24 (38.1%) | 0 (0.0%) |
| 35–39 | 3 (1.2%) | 2 (22.2%) | 48 (34.8%) | 23 (60.5%) | 0 (0.0%) | 29 (46.0%) | 2 (50.0%) |
| Medical insurance * | | | | | | | |
| Association health insurance | 90 (22.0%) | 1 (11.1%) | 56 (40.6%) | 17 (44.7%) | 0 (0.0%) | 16 (25.4%) | 0 (0.0%) |
| Union health insurance | 64 (37.4%) | 1 (11.1%) | 35 (25.4%) | 5 (13.2%) | 0 (0.0%) | 23 (36.5%) | 0 (0.0%) |
| Mutual aid association | 29 (27.9%) | 1 (11.1%) | 17 (12.3%) | 3 (7.9%) | 0 (0.0%) | 8 (12.7%) | 0 (0.0%) |
| National health insurance | 49 (17.4%) | 5 (55.6%) | 15 (10.9%) | 12 (31.6%) | 3 (100.0%) | 11 (17.5%) | 3 (75.0%) |
| National health insurance association | 13 (20.6%) | 1 (11.1%) | 9 (6.5%) | 0 (0.0%) | 0 (0.0%) | 3 (4.8%) | 0 (0.0%) |
| Others | 4 (25.0%) | 0 (0.0%) | 2 (1.4%) | 1 (2.6%) | 0 (0.0%) | 1 (1.6%) | 0 (0.0%) |
| Unknown | 6 (23.1%) | 0 (0.0%) | 4 (2.9%) | 0 (0.0%) | 0 (0.0%) | 1 (1.6%) | 1 (25.0%) |
| Reason for BCS | | | | | | | |
| Information from your municipality or workplace. | 68 (26.7%) | 4 (44.4%) | 31 (22.5%) | 10 (26.3%) | 1 (33.3%) | 22 (34.9%) | 0 (0.0%) |
| Recommendations from your family doctor. | 11 (4.3%) | 0 (0.0%) | 5 (3.6%) | 3 (7.9%) | 0 (0.0%) | 2 (3.2%) | 1 (25.0%) |
| My own health care. | 110 (43.1%) | 2 (22.2%) | 62 (44.9%) | 15 (39.5%) | 0 (0.0%) | 29 (46.0%) | 2 (50.0%) |
| Because I felt I needed to see a doctor. | 48 (18.8%) | 2 (22.2%) | 29 (21.0%) | 7 (18.4%) | 2 (66.7%) | 8 (12.7%) | 0 (0.0%) |
| Because a family member or acquaintance has cancer, and I am concerned. | 13 (5.1%) | 1 (11.1%) | 8 (5.8%) | 2 (5.3%) | 0 (0.0%) | 1 (1.6%) | 1 (25.0%) |
| Others. | 5 (2.0%) | 0 (0.0%) | 3 (2.2%) | 1 (2.6%) | 0 (0.0%) | 1 (1.6%) | 0 (0.0%) |

* Employee insurance mainly includes "Association health insurance (for employees of small and medium-sized companies and their dependents)", "Union health insurance (for employees of large companies and their dependents)", "Mutual aid association (for public employees and their dependents)", and "National health insurance association (for doctors, construction workers and their dependents)". Regional insurance includes "National health insurance (for people who are not covered by employee insurance, such as the self-employed and unemployed)".

A total of 293 (35.9%) respondents were in the 25–29 age group, of which 150 (51.2%) were in regular employment, and 122 (41.6%) had association health insurance. The reasons for not being examined were as follows: 73 (24.9%) did not have a chance to undergo a cancer screening, while 130 (44.4%) had been previously examined for cervical cancer.

Among the 211 (25.9%) in the 30–34 age group, 81 (38.4%) were regularly employed, 66 (31.3%) were housewives, and 94 (44.6%) had association health insurance. The reasons for not having been examined were: "Because I never had a chance to have a chance to have a cancer screening" for 42 (19.9%) and "Participation in cervical cancer screening" for 95 (45.0%).

Among the 215 (26.3%) in the 35–39 age group, 76 (35.4%) were regularly employed, 67 (31.2%) were housewives, and 88 (40.9%) had association health insurance. The reasons for not being examined were as follows: 46 (21.4%) responded that they had not had a chance to be examined, while 80 (37.2%) had been previously examined for cervical cancer.

*3.4. Correlation between BCS and Personal Characteristics*

Correlation coefficients are presented in Supplementary Material Table S2, and Figure 2 was constructed using the coefficients of the correlation between BCS, personal characteristics and perceptions of harms. The correlation coefficients between BCS and perceptions of harms ranged from −0.01 (BCS vs. age) to 0.27 (BCS vs. having regular health checkups). The coefficients of the correlation between age and personal characteristics ranged from −0.33 (age vs. marital status) to 0.05 (age vs. work status). The coefficients of the correlation between marital status and personal characteristics ranged from −0.36 (marital status vs. medical insurance (independent)) to 0.23 (marital status vs. private medical insurance). The coefficients of the correlation between work status and personal characteristics ranged from −0.14 (work status vs. educational background) to 0.60 (work status vs. medical insurance (dependent)).

**Table 4.** Psychological and personal characteristics affecting non-participation in BCS.

| Characteristic | Total (n = 816) | Age 20–24 (n = 97) | 25–29 (n = 293) | 30–34 (n = 211) | 35–39 (n = 215) |
|---|---|---|---|---|---|
| **Work status** | | | | | |
| Self-employed | 26 (3.2%) | 1 (1.0%) | 6 (2.1%) | 7 (3.3%) | 12 (5.6%) |
| Regular employment | 340 (41.7%) | 33 (34.0%) | 150 (51.2%) | 81 (38.4%) | 76 (35.4%) |
| Part-time job | 152 (18.6%) | 14 (14.4%) | 49 (16.7%) | 44 (20.9%) | 45 (20.9%) |
| Student | 46 (5.6%) | 43 (44.3%) | 3 (1.0%) | 0 (0.0%) | 0 (0.0%) |
| Housewife | 205 (25.1%) | 3 (3.1%) | 69 (23.6%) | 66 (31.3%) | 67 (31.2%) |
| Unemployed | 47 (5.8%) | 3 (3.1%) | 16 (5.5%) | 13 (6.2%) | 15 (7.0%) |
| **Medical insurance \*** | | | | | |
| Association health insurance | 319 (78.0%) | 15 (15.5%) | 122 (41.6%) | 94 (44.6%) | 88 (40.9%) |
| Union health insurance | 107 (62.6%) | 12 (12.4%) | 36 (12.3%) | 30 (14.2%) | 29 (13.5%) |
| Mutual aid association | 75 (72.1%) | 10 (10.3%) | 29 (9.9%) | 16 (7.6%) | 20 (9.3%) |
| National health insurance | 233 (82.6%) | 41 (42.3%) | 75 (25.6%) | 54 (25.6%) | 63 (29.3%) |
| National health insurance association | 50 (79.4%) | 8 (8.3%) | 20 (6.8%) | 10 (4.7%) | 12 (5.6%) |
| Unknown | 20 (76.9%) | 8 (8.3%) | 7 (2.4%) | 5 (2.4%) | 0 (0.0%) |
| Others | 12 (75.0%) | 3 (3.1%) | 4 (1.4%) | 2 (1.0%) | 3 (1.4%) |
| **Reason for not participating in BCS** | | | | | |
| Busy. | 76 (9.3%) | 8 (8.3%) | 22 (7.5%) | 23 (10.9%) | 23 (10.7%) |
| I'm healthy. | 38 (4.7%) | 3 (3.1%) | 9 (3.1%) | 12 (5.7%) | 14 (6.5%) |
| I am anxious about the results. | 27 (3.3%) | 0 (0.0%) | 14 (4.8%) | 7 (3.3%) | 6 (2.8%) |
| Because I did not know about cancer screening. | 15 (1.8%) | 5 (5.2%) | 8 (2.7%) | 2 (1.0%) | 0 (0.0%) |
| Because I never had a chance to have a cancer screening. | 184 (22.6%) | 23 (23.7%) | 73 (24.9%) | 42 (19.9%) | 46 (21.4%) |
| Because I forgot to take the test. | 57 (7.0%) | 7 (7.2%) | 12 (4.1%) | 14 (6.6%) | 24 (11.2%) |
| I don't think I am old enough to have a checkup. | 58 (7.1%) | 32 (33.0%) | 16 (5.5%) | 5 (2.4%) | 5 (2.3%) |
| Participation in cervical cancer screening. | 321 (39.3%) | 16 (16.5%) | 130 (44.4%) | 95 (45.0%) | 80 (37.2%) |
| Others. | 40 (4.9%) | 3 (3.1%) | 9 (3.1%) | 11 (5.2%) | 17 (7.9%) |

\* Employee insurance mainly includes "Association health insurance (for employees of small and medium-sized companies and their dependents)", "Union health insurance (for employees of large companies and their dependents)", "Mutual aid association (for public employees and their dependents)", and "National health insurance association (for doctors, construction workers and their dependents)". Regional insurance includes "National health insurance (for people who are not covered by employee insurance, such as the self-employed and unemployed)".

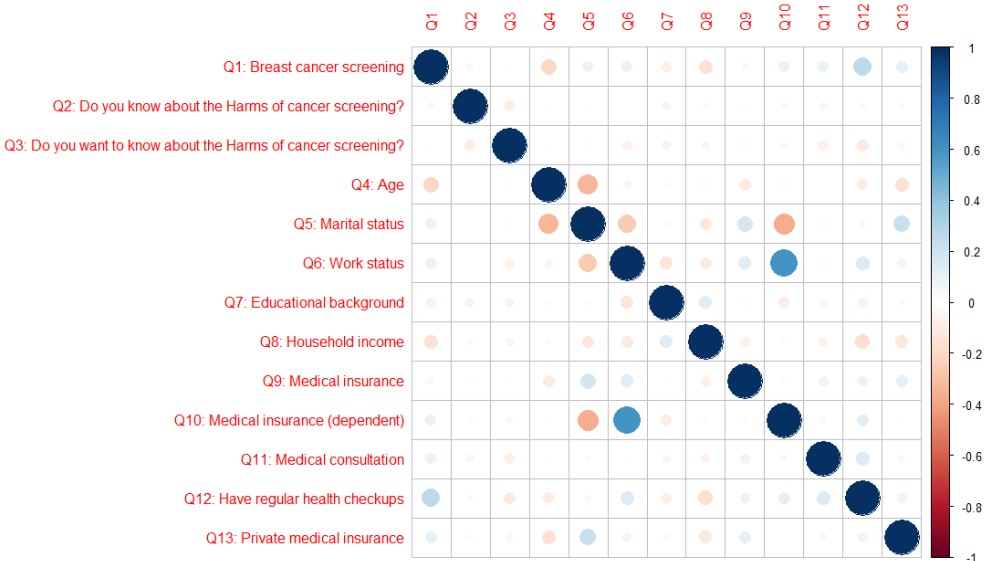

**Figure 2.** Coefficient of the correlation between BCS and personal attributes. Blue circles represent positive correlations (0.0 to 1.0) between variables, and red circles represent negative correlations (−1.0 to 0.0) between variables. The darker the color, the stronger the correlation.

The correlation between BCS and personal characteristics was r < 0.7 (|r| < 0.7). There were no multicollinearity problems.

### 3.5. Psychological and Personal Characteristics Affecting Participation in BCS

A logistic analysis was performed excluding medical insurance (dependent variable) from the results of the correlation analysis and using BCS (1: BCS participation, 0: BCS non-participation) as the dependent variable.

Table 5 shows the results obtained for the psychological characteristics of the screened group based on the HBM; the odds ratios were significantly lower for "burden to participation before cancer screening", which was 0.44 (95% CI, 0.32–0.60) ($p < 0.001$). Among the personal characteristics significantly associated with screening behaviors, aged 25–29 was 0.33 (95% CI, 0.21–0.51) ($p < 0.001$); regular employment was 6.96 (95% CI, 1.78–27.14) ($p = 0.005$); part-time job was 5.38 (95% CI, 1.34–21.59) ($p = 0.018$); housewife was 5.16 (95% CI, 1.28–20.91) ($p = 0.021$); and undergoing regular health checkups was 1.80 (95% CI, 1.23–2.63) ($p = 0.003$). On the other hand, regarding knowledge of the harms of cancer screening, the value of those who did not want to know was 5.34 (95% CI, 1.31–21.37) ($p = 0.019$).

**Table 5.** Psychological and personal characteristics affecting participation of BCS.

| Parameter | OR | 95%CI | *p*-Value |
|---|---|---|---|
| **Do you want to know about the harms of cancer screening?** | | | |
| I want to know very much. | Ref. | —— | —— |
| I don't want to know at all. | 5.34 | 1.31–21.73 | 0.019 |
| I don't want to know so much. | 1.40 | 0.39–5.06 | 0.606 |
| I can't say either. | 1.78 | 1.02–3.11 | 0.042 |
| I want to know. | 1.24 | 0.79–1.95 | 0.345 |
| **Age** | | | |
| 35–39 | Ref. | —— | —— |
| 20–24 | 0.49 | 0.21-1.17 | 0.109 |
| 25–29 | 0.33 | 0.21–0.51 | <0.001 |
| 30–34 | 0.77 | 0.52–1.15 | 0.207 |
| **Work status** | | | |
| Unemployed | Ref. | —— | —— |
| Self-employed | 5.44 | 1.14–25.97 | 0.034 |
| Regular employment | 6.96 | 1.78–27.14 | 0.005 |
| Part-time job | 5.38 | 1.34–21.59 | 0.018 |
| Student | 1.62 | 0.25–10.60 | 0.617 |
| Housewife | 5.16 | 1.28–20.91 | 0.021 |
| **Have regular health checkups** | | | |
| No | Ref. | —— | —— |
| Yes | 1.80 | 1.23–2.63 | 0.003 |
| **HBM** | | | |
| Susceptibility to cancer | 1.00 | 0.82–1.22 | 0.984 |
| Seriousness of cancer | 1.04 | 0.80–1.35 | 0.777 |
| Benefits of cancer screening | 1.13 | 0.86–1.47 | 0.380 |
| Burden to participation before cancer screening | 0.44 | 0.32–0.60 | <0.001 |
| Burden to participation at the time of cancer screening | 1.16 | 0.91–1.47 | 0.235 |
| Importance of cancer screening | 0.82 | 0.61–1.11 | 0.204 |
| Cues to participation in screening | 1.13 | 0.91–1.40 | 0.259 |

### 3.6. Psychological and Personal Characteristics Affecting Participation in Knowledge of the Harms of Cancer Screening

Table 6 shows the results obtained for the psychological characteristics of the screened group based on the HBM. The odds ratios were significantly higher for "susceptibility to cancer", which was 1.20 (95% CI, 1.00–1.43) ($p = 0.045$), and "benefits of cancer screening",

which was 1.30 (95% CI, 1.06–1.60) (*p* = 0.013), whereas the odds ratio was significantly lower for "importance of cancer screening", which was 0.61 (95% CI, 0.48–0.78) (*p* < 0.001). Among the personal characteristics significantly associated with screening behaviors, regular employment was 1.89 (95% CI, 1.01–3.54) (*p* = 0.046), and part-time job was 1.98 (95% CI, 1.00–3.93) (*p* = 0.050).

**Table 6.** Psychological and personal characteristics affecting the knowledge of harms regarding BCS.

| Parameter | OR | 95%CI | *p*-Value |
|---|---|---|---|
| Work status | | | |
| Unemployed | Ref. | — | — |
| Self-employed | 1.79 | 0.66–4.89 | 0.253 |
| Regular employment | 1.89 | 1.01–3.54 | 0.046 |
| Part-time job | 1.98 | 1.00–3.93 | 0.050 |
| Student | 1.71 | 0.60–4.85 | 0.315 |
| Housewife | 1.60 | 0.83–3.06 | 0.160 |
| HBM | | | |
| Susceptibility to cancer | 1.20 | 1.00–1.43 | 0.045 |
| Seriousness of cancer | 0.99 | 0.80–1.21 | 0.892 |
| Benefits of cancer screening | 1.30 | 1.06–1.60 | 0.013 |
| Burden to participation before cancer screening | 1.05 | 0.79–1.38 | 0.744 |
| Burden to participation at the time of cancer screening | 1.17 | 0.94–1.44 | 0.161 |
| Importance of cancer screening | 0.61 | 0.48–0.78 | <0.001 |
| Cues to participation in screening | 1.02 | 0.84–1.23 | 0.839 |

## 4. Discussion

To our knowledge, this is the largest study on youth BCS and its associated harms reported in the past 20 years. Our study results indicate that there is an association between undergoing BCS and interest in its harms. In particular, we believe that it is possible to adjust the balance between the benefits and harms of BCS among young adults who are in regular and part-time employment depending on whether or not they are interested in the harms of BCS.

The $\chi^2$ test showed that there were significant differences in age, marital status, work status, educational background, household income, medical insurance, medical consultation, having regular health checkups, and private medical insurance. This was consistent with the results of previous studies [27–30].

About one-third of the younger age groups were examined by BCS. The examinees were characterized as being in the 30–39 age group. They were regularly employed with association health insurance and union health insurance, part-timers, or housewives who received BCS for self-health reasons. We attribute this to younger women's interest in BCS [33] and their earlier life stage, which may lead them to ignore information about BC and risk-reduction behaviors because they have less involvement and experience with the healthcare system than older women [34]. On the other hand, the reason for not being examined was the perception that BCS was less important than cervical cancer screening. Breast awareness education is needed for this group.

Breast awareness comprises the following four practices: (1) knowing the condition of your breasts (look, touch, and feel—breast check); (2) being aware of breast changes (lumps, skin indentations, and bloody nipple discharge); (3) consulting your doctor as soon as you notice a change; and (4) having a breast cancer screening once every two years after the age of 40. Breast awareness is also a key concept in the development of breast cancer prevention. Breast awareness was also recommended by the World Health Organization (WHO) as an effective medical policy for breast cancer control that should be implemented worldwide, regardless of funding availability.

As a part of breast awareness education in Japan, group screening is strongly recommended for women over 40. This recommendation could invite health concerns for women

aged 30–39. However, it is thought to be most important for examinees to take action after considering the balance of benefits and harms of medical checkups by themselves, while improving their awareness of BC and BCS rather than being anxious.

The logistic analysis showed that among the 25–29 age group, the more they felt the "burden to participation before cancer screening", the more likely they were not to participate in BCS. This was because many 25–29-year-old respondents were regularly employed with an association health plan. These women have limited time when they receive regular health checkups and consider the checkups in order of importance, from cervical cancer to BC; therefore, they think that they have few opportunities to receive BCS. On the other hand, we considered that regular employees, those with a part-time job, and housewives receive BCS during regular health checkups (occupational health checkups) depending on the insurance type of the company where they or their husbands work.

The workplace-based screening was in the company benefits category, and the content, appropriate age, and methods of checkup varied; additionally, the accuracy control of checkups is poor, increasing the harms of BCS. Therefore, we believe that greater accuracy control of job area checkups is necessary.

Screening accuracy management has been poor because of the characteristics of the workplace-based screening provided by the provider, in which the examinee can receive the screening even if he or she does not have knowledge of the disease. Thus, to solve the issue of frequent harms, the national and local governments should cooperate in disseminating and raising awareness of correct cancer screening knowledge. Second, because the harms of screening may outweigh the benefits, insurers and employers should correctly understand and explain the benefits (e.g., reduction in mortality) and harms (e.g., false positive, overdiagnosis, complications from testing) to those undergoing screening [35]. Finally, because various tools are used to acquire knowledge about cancer screening, but knowledge acquisition varies among individuals [36], it is necessary for healthcare professionals (occupational physicians) to educate examinees about smart searches and reliable websites [37,38] regarding screening.

On the other hand, an important finding was that among those with a high interest regarding the harms of undergoing BCS and the associated harms of screening (typically younger age regular employees or those with a part-time job), the higher the interest in harm, the more correct their perceptions of "susceptibility to cancer", "benefits of cancer screening", and "importance of cancer screening". In the future, as knowledge of the harm specific to younger age groups becomes more widespread, regular employees and parttime workers are more likely not to undergo BCS. In addition, young adults may be unaware of their family health history [39,40]; therefore, they may not be aware of their own breast abnormalities or may delay seeing a doctor, even if they are aware [41,42]. As such, there is a need to promote BSE (breast self-examination) to reduce the harmful effects of BCS (radiation exposure [5], false positives [6], and overdiagnosis [11]) in young adults.

In Japan, young adults are a more important target group for breast health promotion than other age groups [43,44]. In addition, breast self-examination (BSE) should be performed starting at age 20; therefore, it is necessary for young women to understand their breasts and be aware of changes [44].

BSE is an effective method to detect the changes associated with BC [45–47], and as a way to raise awareness of breast health, it is an easy and cost-effective procedure; however, due to a lack of knowledge regarding BSE [46–48], women in many countries do not regularly perform self-examination breast checks [43–45,49]. Therefore, we believe that BSE education is needed.

This study shows that the implementation of "Breast Awareness", "Accuracy Management of Workplace Screening", and "Breast Self-Examination" will increase awareness of the downsides of screening, deepen understanding of cancer screening, and enable people to judge for themselves the balance of the pros and cons of BCS before attending screening, as well as highlight the importance of taking action to undergo screening after judging the balance of the pros and cons of BCS.

The limitations of this study are: First, we selected respondents through an online source, and it is not possible to confirm that 100% of them are Japanese women. Second, the samples regarding the harms of screening needed to be increased. Finally, this study only asked the respondents whether they thought they knew the harms or were interested in knowing the harms; it did not assess their actual knowledge of harms and the correctness of that knowledge, so the harms need to be quantified and explored in depth. In addition, the Japanese female labor force is characterized by a bias in employment status by age, so it is necessary to consider how to adjust for this.

The health belief questions were not specific to BCS, but for women's cancer screening in general. The attitudes may differ for the screening of BC vs. cervical cancer, so in the future, we should survey using a behavior change questionnaire specifically designed for BCS.

## 5. Conclusions

This study identified: (i) the influencing factors of BCS participation among young people and their relationship to perceptions regarding the harms of screening; (ii) low awareness of the harms of BCS and a high level of concern among young people; (iii) the large number of young people who receive BCS during their workplace-based screening; and (iv) that the greater the awareness of the harms of BCS, the less likely the examinees were to undergo BCS at their workplace-based screening. In addition, breast awareness education and the dissemination of breast self-examination information are needed to reduce the harms of BCS among younger age groups.

On the other hand, because many people undergo BCS in workplace health checkups, it is necessary to control the accuracy of workplace health checkups. Specifically, it is necessary to promote BCS for appropriate age groups and explain the harmful effects of BCS with risk stratification (young, middle-aged, and older adults), in addition to explaining the testing methods suitable for young adults.

**Supplementary Materials:** The following supporting information can be downloaded at: https://www.mdpi.com/article/10.3390/curroncol30020161/s1, Table S1: All the personal characteristics used in this study. Table S2: Correlation matrix among BCS and personal attributes. Supplementary Material S1: Distinction between medical insurance and private insurance.

**Author Contributions:** Conceptualization, Z.C. and H.K.; data collection and processing, Z.C. and M.T.; analysis, Z.C.; writing—original draft preparation, Z.C.; writing—review and editing, H.K., Y.C., M.M.R., S.Y., Y.L. and M.K.; interpreter check, M.M.R. All authors have read and agreed to the published version of the manuscript.

**Funding:** This research was funded by the Grants-in-Aid for Scientific Research Program (KAKENHI), Japan, grant number 15H04751 (to M.K.).

**Institutional Review Board Statement:** Not applicable.

**Informed Consent Statement:** This study extracted data from the 2018 online survey (Approval No. E-1081-1); therefore, ICS was not required.

**Data Availability Statement:** The data are not publicly available due to confidentiality reasons.

**Conflicts of Interest:** The authors declare no conflict of interest. The funders had no role in the study design and analysis, decision to publish, or manuscript preparation.

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
