# Peer review of "Breast Cancer Screening and Perceptions of Harm among Young Adults in Japan: Results of a Cross-Sectional Online Survey"

_curroncol, doi:10.3390/curroncol30020161_

Round 1
Reviewer 1 Report
1. It seems to me that there is no need to give table 1, since columns 2 and 3 are the same in all cases. The first column can be given in the text.
2. Table 4 is completely unreadable.
3. Nevertheless, cases of breast cancer under the age of 30 are very rare, at the age of 30-39 it is more common, but of course this frequency of occurrence is not enough to include this age group in the mass screening program (mammography). Wouldn't drawing attention to the need for screening increase anxiety in this age group? Will the frequency of self-referral to the doctor increase, which will include additional examinations that will increase the risk of cancer later on?
4. It seems to me that it would be more informative to compare the results of the survey in the group under 40 and over 40. And so the authors got the result that women under the age of 30 are less likely to participate in breast cancer screening programs than women aged 30-39, but this is quite normal. Women with family hereditary breast cancer will be more conscious, and this is also natural. What fundamentally new have we learned from the results obtained?
Author Response
Point 1: It seems to me that there is no need to give table 1, since columns 2 and 3 are the same in all cases. The first column can be given in the text.
Response 1: Thank you for your suggestion. The original Table 1 has been removed with text explaining the source of the data.
We have added and revised the text as follows: “The personal characteristics, BCS participation status, reasons for non-participating in BCS, knowledge of the harms of cancer screening, and HBM used in this study were all derived from the "2018 Attitude Survey on Factors Influencing Women's Cancer Screening Behavior." (Page 3, line 105-108). In addition, the Table numbers have also been updated.
Point 2: Table 4 is completely unreadable.
Response 2: Thank you for your comments. We have revised and modified Table 3 (original Table 4) to make it readable (Page 8, Table 3. Status of participation in BCS).
Point 3: Nevertheless, cases of breast cancer under the age of 30 are very rare, at the age of 30-39 it is more common, but of course this frequency of occurrence is not enough to include this age group in the mass screening program (mammography). Wouldn't drawing attention to the need for screening increase anxiety in this age group? Will the frequency of self-referral to the doctor increase, which will include additional examinations that will increase the risk of cancer later on?
Response 3: Thank you for these important queries. The results of the analysis of this study, Table 3, "My own health care," and Table 4, "Because I never had a chance to have a cancer screening" and "Participation in cervical cancer screening," indicate that young adults are more health conscious about BCS and are more likely to have a cancer screening if given the opportunity. As you mentioned, we think that strongly encouraging group screening as an opportunity for young people to receive medical examinations may invite health concerns. This age group is more likely to develop overdiagnosis and false positives by screening for BCS. In accordance with the results of Table 5 and Table 6 of this study, it recommends a high level of awareness of BCS accompanied by increased awareness of disadvantages and a well-informed screening process. From the results of all analyses, young people need to implement "Breast Awareness", "Accuracy Management of workplace screening", and "Breast Self-Examination" instead of receiving MG.
We have added and revised the text as follows: “Part of breast awareness education in Japan strongly recommends group screening for women over 40. This recommendation could invite health concerns for women aged 30-39. However, it is thought to be most important for examinees to take action after considering the balance of benefits and harms of medical checkups by themselves, while im-proving their awareness of BC and BCS rather than being anxious." (Page 13, line 315-319).
Point 4: It seems to me that it would be more informative to compare the results of the survey in the group under 40 and over 40. And so the authors got the result that women under the age of 30 are less likely to participate in breast cancer screening programs than women aged 30-39, but this is quite normal. Women with family hereditary breast cancer will be more conscious, and this is also natural. What fundamentally new have we learned from the results obtained?
Response 4: Thank you for these valuable remarks. As you pointed out, it might be instructive to compare the findings for the groups under 40 and over 40. Selection of the target to be approached requires segmentation of the target population. As an exploratory study for future research, this study will also be used to generate hypotheses. It will initially separate the age groups into 20-30 years and 40-60 years to determine the characteristics of breast cancer screening, and then quantify the harms and separate the groups into under 40 and over 40 years of age. From Table 3 and Table 4, we can see that the alternatives of "breast awareness," "accuracy management of workplace screening," and "breast self-examination" are considered for young people who have a strong health consciousness and tend to undergo BCS screening regardless of understanding the benefits or harms of BCS when they have the opportunity to undergo BCS screening. Finally, from Table 6, increasing awareness of the harms of screening increases understanding of cancer screening and makes it a behavior to not undergo screening on their own. With this result, the achievement from MG to the reduction of falsifiability was a new finding from this study.
We have added and revised the text as follows: “This study shows that the implementation of "Breast Awareness", "Accuracy Management of Workplace Screening", and "Breast Self-Examination" will increase awareness of the downsides of screening, deepen understanding of cancer screening, and enable people to judge for themselves the balance of the pros and cons of BCS before attending screening, and the importance of taking action to undergo screening after judging the balance of pros and cons of BCS." (Page 14, line 366-371).
Reviewer 2 Report
Thank you for the opportunity to review the study. Please see my comments below.
1. Lines 41-42. Mammography or visual inspection cannot treat BC. Please revise.
2 More information about how the online survey is conducted is needed. The method section should describe how participants are selected and administered the surveys, the response rates etc.
3. Since all items are from the same survey of the same year, Table 1 could be simplified without the columns of period and Source.
4. It’s not clear if the logistic regression results reported in table 6 are for risk of having BCS or not having BCS. This needs to make clear both in the method section and related results and tables.
5. Please describe how HBM is used as a theoretical framework to identify factors.
6. The specific aims of this study need to be clearly stated in the introduction.
7. The survey asks whether a participant knows the harms and the attitudes towards knowing the harms of BCS. But it does assess their actual knowledge of harms. It is unknown whether participants have a correct understanding of harms from BCS. And the harm questions are also not BC specific.
8. Lines 130 and 135. Cronbach's alpha is a measure of internal consistency, that is, how closely related a set of items are as a group. It is not clear why higher score indicates that participation in screening was perceived more beneficial. Please clarify.
9. It is not clear how missing data was checked. To have 0% missing in an online survey seems to be too good to be true. Did the authors only check missing on BCS variable?
10. It is not clear why the authors chose |r|<0.5 as a threshold for variables included in the logistic regression. A correlation of |r|>=0.5 is considered large correlation. The variables that have high correlation with receipt of BCS may be different from those associated with presence of interest to know about harm. This should be tested separately for receipt of BCS and harms. In line 150, it is not clear what the p<0.05 mean in that sentence.
11. It’s not clear where the authors obtained the information on receipt of BCS. Table 1 does not have that information.
12. The health belief questions were not specific to BCS, but for women cancer screening in general. The attitudes may be different for screening of BC vs. cervical cancer.
13. Lines 164-167. The difference between medical insurance and private insurance is unclear. It seems private insurance and medical insurance overlaps and one should be included but not two.
14. Table 3. Some are reporting row %. Should report column % for all variables.
15. Table 3. Answers are the same for both know and what to know questions.
16. The wanting to know question asks whether a person wants to know, not whether a person is concerned about harm. It’s possible that after learning of the harms, a person may not be concerned.
17. Table 4, the working status subcategories have very low numbers. Consider delete the table since the estimates are likely unreliable using such small samples.
18. Suggest using consistent term to describe receipt of BCS. Variables terms were used throughout the test: BCS non-attendees, participation in BCS, receipt of BCS, presence/absence of a BCS visit, examinees etc.
19. lines 236-7. It is not clear how the authors obtained |r|<0.7 for correlation of BCS with personal characteristics. The interpretation of this estimate is also not clear. The exact value should be reported. r<0.7 could be 0.1 or 0.65. If this is reporting on |r| across all characteristics, then a range of |r| for the variables should be reported.
20. It states that Figure 2 reports correlation coefficients. But the line above says it is logistic analysis. Also footnote should be included under Figure 2 to explain how to interpret the graph and the meaning and significance of the various colors and circles.
21. Decision to get BCS should be a personal decisions based on person risk factors, and a balance of benefits vs. harms. It seems the authors were only emphasizing the harms in this article.
Author Response
Point 1: Lines 41-42. Mammography or visual inspection cannot treat BC. Please revise.
Response 1: We thank the reviewer for this insightful suggestion. We have revised the text as follows: " Prior studies have shown that mammography (MG) alone or in combination with visual inspection is effective in reducing BC mortality through the early detection of BC [3-4]." (page 1, lines 31-33).
Point 2: More information about how the online survey is conducted is needed. The method section should describe how participants are selected and administered the surveys, the response rates etc.
Response 2: Thank you very much for your valuable comments. We have added more information in the manuscript as follows: “The online survey title was "2018 Attitude Survey on Factors Influencing Women's Cancer Screening Behavior." The survey used "registered monitors" owned by an online research company (Hiroshima, Japan). Samples were drawn according to the percentage of the population (see 2015 Census) in each age category (20s, 30s, 40s, 50s, and 60s). The survey was stopped when the target sample for each category was reached. A sample of approximately 3,000 persons was deemed adequate.” (Page 3, line 98-103).
Point 3: Since all items are from the same survey of the same year, Table 1 could be simplified without the columns of period and Source.
Response 3: Thank you for your helpful suggestion. The original Table 1 has been removed with text explaining the source of the data.
We have added and revised the text as follows: “The personal characteristics, BCS participation status, reasons for non-participating in BCS, knowledge of the harms of cancer screening, and HBM used in this study were all derived from the "2018 Attitude Survey on Factors Influencing Women's Cancer Screening Behavior". (Page 3, line 105-108). In addition, the Table numbers have also been updated.
Point 4: It’s not clear if the logistic regression results reported in table 6 are for risk of having BCS or not having BCS. This needs to make clear both in the method section and related results and tables.
Response 4: Many thanks for your thoughtful comments. We have added and revised the text as follows: “The flow of logistic analysis in our study was as follows: (i) We conducted correlation analysis based on the results of the χ2 test (p<0.05) and the perception of harms. (ii) We conducted a logistic regression analysis with the BCS (1: Participation in BCS, 0: Non-participation in BCS) as the dependent variable, and the variable with |r|<0.5 obtained from the correlation analysis and the seven factors of HBM as explanatory variables." (Page 4, line 152-157).
“Excluding medical insurance (dependent), we conducted a logistic analysis with BCS (1: Participation in BCS, 0: Non-participation in BCS) as the dependent variable." (Page 11, line 262-263).
Point 5: Please describe how HBM is used as a theoretical framework to identify factors.
Response 5: Thank you for your question. It is important for this study not only to identify factors, but also to observe the associations between factors and behavior. Therefore, we used the theoretical framework of HBM to identify the factors.
Point 6: The specific aims of this study need to be clearly stated in the introduction.
Response 6: Thank you for your comments. The objectives of this study were (1) To identify the factors that influence young people's BCS screening. (2) To identify factors that influence the perception of harmfulness regarding BCS among young people.
We have revised the explanation as follows: “Therefore, this study uses the HBM as a theoretical framework with the aims (i) to identify factors that influence BCS and (ii) to identify factors that influence the perception of harm about BCS in young adults. We also fill a gap in the literature regarding the up-take of BCS among young adults according to the interest in BCS harms.” (Page 2, line 89-92).
Point 7: The survey asks whether a participant knows the harms and the attitudes towards knowing the harms of BCS. But it does assess their actual knowledge of harms. It is unknown whether participants have a correct understanding of harms from BCS. And the harm questions are also not BC specific.
Response 7: Thanks for your indepth opinion. As you indicated, there are limits to the specificity of the disadvantage and the ability to make an objective determination. First, the purpose of this study is to understand whether young people are receiving BCS, knowing that the harms of BCS exist. It was also to ascertain the association between the perception of BCS disadvantage and BCS screening behavior. Verifying the specifics of harms are difficult; knowledge assessment without knowing whether or not people are aware of the existence of harms in prior surveys involves the risk that the survey will fail. If no one believes that they know of any disadvantage, no answer will be given. This may not be good for research evolution. Therefore, this study first asked about perceptions of subjective disadvantage. Based on the results of the study to date, we plan to quantify the harms and deepen the study in the future. The ultimate goal of the study was not to know the harms of breast and cervical cancer alone, but to understand the overall harms and to generate screening behavior. We have added and revised the text as follows: “Finally, the harms of BCS, such as decreases in life expectancy and longevity, are subjective responses, and the harms need to be quantified and explored in depth. In addition, the Japanese female labor force is characterized by a bias in employment status by age, so it is necessary to consider how to adjust for this."(Page 14, line 374-378).
Point 8: Lines 130 and 135. Cronbach's alpha is a measure of internal consistency, that is, how closely related a set of items are as a group. It is not clear why higher score indicates that participation in screening was perceived more beneficial. Please clarify.
Response 8: Many thanks for your useful comment. We have revised and added the text as follows: “The larger the Cronbach's α coefficient, the more reasonably the variables constructed on the basis of the factors obtained are considered to be represented.” (Page 3, line 143-145).
Point 9: It is not clear how missing data was checked. To have 0% missing in an online survey seems to be too good to be true. Did the authors only check missing on BCS variable?
Response 9: The study checked for missing status on all variables. The reason for the zero deficiencies was to reduce the burden on respondents, who participated in the design and modification of the questionnaire from time to time, starting with the creation of the online questionnaire. In order to ensure the reliability of the respondents' responses, we still actually operated the online survey and measured the minimum time required for the actual survey. Then, we set the questionnaires whose response time was less than the minimum time as meaningless data.
Point 10: It is not clear why the authors chose |r|<0.5 as a threshold for variables included in the logistic regression. A correlation of |r|>=0.5 is considered large correlation. The variables that have high correlation with receipt of BCS may be different from those associated with presence of interest to know about harm. This should be tested separately for receipt of BCS and harms. In line 150, it is not clear what the p<0.05 mean in that sentence.
Response 10: Thank you for your questions. In general, regression models often suffer from multicollinearity problems. There is now an index VIF (Variance Inflation Factor) that measures multicollinearity. In general, if the VIF statistic is greater than 10, there is a possibility that multicolinearity exists. Therefore, it is necessary to check if the VIF is greater than 10. The VIF can be calculated using the correlation coefficient between explanatory variables with the following formula.
VIF=1/(1-r^2 )
※ris the correlation coefficient.
※For example, r =0.1 (VIF=1.01), r =0.5 (VIF=1.33), r =0.9 (VIF=5.26)
The higher the correlation coefficient, the greater the VIF. In addition, the correlation coefficient is the coefficient of the correlation between r ranging from -1 to 1, where r.0.0~0.2(Hardly any correlation), 0.2~0.5(Somewhat correlated), 0.5~0.7(Quite a correlation), 0.7~1.0(Strong correlation). The study was summarized within 10 VIFs by selecting variables that were somewhat correlated |r|<0.5 among the variables.
To clarify the association between breast cancer screening and disadvantage, this study introduced the BCS influence factor as an explanatory variable for harms.
Page 5, line158 p<0.05 is a significant result obtained from logistic regression.
Point 11: It’s not clear where the authors obtained the information on receipt of BCS. Table 1 does not have that information.
Response 11: Thank you very much for your valuable comment. We have added the text as follows: “The online survey title was "2018 Attitude Survey on Factors Influencing Women's Cancer Screening Behavior." The survey used "registered monitors" owned by an online research company (Hiroshima, Japan). Samples were drawn according to the percentage of the population (see 2015 Census) in each age category (20s, 30s, 40s, 50s, and 60s). The survey was stopped when the target sample for each category was reached. A sample of approximately 3,000 persons was deemed adequate.” (Page 3, line 98-103).
Point 12: The health belief questions were not specific to BCS, but for women cancer screening in general. The attitudes may be different for screening of BC vs. cervical cancer.
Response 12: As you pointed out, a detailed investigation would be different. However, in this study, we first considered it largely as cancer specific to women as an exploratory study for hypothesis generation.
Point 13: Lines 164-167. The difference between medical insurance and private insurance is unclear. It seems private insurance and medical insurance overlaps and one should be included but not two.
Response 13: Thank you for your questions. Medical insurance and private insurance in this study cannot be merged. This section describes the health situation in Japan. The reasons for this are: ①Medical insurance is Japan's public medical insurance. Public health insurance is compulsory for all citizens. In other words, it is a system in which citizens support each other so that everyone can receive medical care with peace of mind. Private insurance, on the other hand, provides hospitalization and surgical benefits as the main policy, although the coverage varies by the insurance company. The major distinction between the two insurance policies is shown in Table 1. ②The questions for medical insurance and private insurance are different and cannot be merged. It is preferable to describe it as a characteristic of the health care system in the text. However, it would be redundant and should be explained only in the peer review reply.
Table 1. Distinction between medical insurance and private insurance
|
|
Medical Insurance |
Private Insurance |
|
Eligibility |
Mandatory Enrollment |
Voluntary Enrollment |
|
Purpose of Membership |
Stability of people's lives |
Co-pay coverage |
|
Insurance Premium |
Depends on income |
Varies by age, gender, and coverage |
|
Benefit |
Benefit in Kind |
Receipt of insurance proceeds |
|
Types and Systems |
Health insurance, high-cost medical care, etc. |
Medical insurance, cancer insurance, etc. |
Point 14: Table 3. Some are reporting row %. Should report column % for all variables.
Response 14: Thank you for your helpful suggestions. Corrected % in Table 2 (original Table 3 ). (Page 6, Table 2. Associations between BCS behaviors and personal characteristics/harms).
We have added the text as follows: "Observing the number of respondents by item showed that the numbers for an awareness of harms were 12 (24.5%) and 37 (75.5%) for those who did or did not receive BCS, respectively. On the other hand, 194 (24.2%) and 607 (75.8%) of the examined and unexamined respondents, respectively, were concerned about the harms of BCS.” (Page 5, line 177-180).
Point 15: Table 3. Answers are the same for both know and what to know questions.
Response 15: Thank you for your remarks. Table 2 ( original Table 3 ) questions were modified.(Page 6, Table 2. Associations between BCS behaviors and personal characteristics/harms).
Point 16: The wanting to know question asks whether a person wants to know, not whether a person is concerned about harm. It’s possible that after learning of the harms, a person may not be concerned.
Response 16: Thank you for your questions. Considering only the answer of wanting to know, it is judged as you have indicated. Table 3, the results of the analysis of this study, shows the reasons for seeing a doctor「My own health care」, Table4, Reasons for not seeing a doctor「Because I never had a chance to have a cancer screening」, and「Participation in cervical cancer screening」. Younger adults are more health-conscious about BCS and are more likely to receive a checkup if they have the opportunity. For younger adults, we believe that recommending that they do not take the BCS may lead to health insecurity. Yet, this age group is more likely to develop overdiagnosis and false positives by screening for BCS. This study recommends a high awareness of BCS accompanied by an increased awareness of harms and a well understood screening according to the results of Table 5 and Table 6. Yet, from Table 3, instead of receiving MG, it is important to implement "Breast Awareness", "Accuracy Management of Workplace Screening", and "Breast Self-Examination". We have discussed the results of our analysis in this comprehensive review.
Point 17: Table 4, the working status subcategories have very low numbers. Consider delete the table since the estimates are likely unreliable using such small samples.
Response 17: Thank you for your comments. The bias in the number of n in the subcategory of employment status in Table 3 (original Table 4) is influenced by the life stage of women in Japan. This is indicated by the M-shaped curve of the employment status of women in Japan in relation to age, and national policies to improve the situation are being considered. Increasing the number of subjects will not eliminate this bias. However, those who have been employed since Table 3 (original Table 4) (with Association Health Insurance and Union Health Insurance) have received BCS for self-health. On the other hand, we have identified a trend that housewives (with National Health Insurance or their husband's company insurance) are visiting the BCS for community/workplace notices or for self-health reasons. With this finding, the study recommended Breast Awareness, Accuracy Management of Workplace Screening, and Breast Self-Examination as alternatives to receiving MG. Working status is important for this study and can contribute to hypothesis generation. These are added to the limitations of the study.
Point 18: Suggest using consistent term to describe receipt of BCS. Variables terms were used throughout the test: BCS non-attendees, participation in BCS, receipt of BCS, presence/absence of a BCS visit, examinees etc.
Response 18: Thank you for your careful review of our paper. We have added the text as follows: “We also fill a gap in the literature regarding the uptake of BCS among young adults according to the interest in BCS harms.” (Page 2, line 91-92).
“The reasons for non-participating in BCS [31] options were busy, healthy, anxiety about the results, lack of awareness about cancer screening, lack of opportunity to have a cancer screening, forgetting to take the test, self-perception of not being old enough to have a checkup, participation in cervical cancer screening, etc.).” (Page 3, line 126-129).
“To examine the association between the BCS and personal characteristics and perceptions of harms, we used the χ2 test.” (Page 4, line 148-150).
“Next, we performed basic tabulations to examine the factors that contributing to participation or non-participation in BCS.” (Page 4, line 150-151).
“We analyzed the association between the BCS and individual characteristics and harms: age, marital status, work status, educational background, household income, medical insurance, medical insurance (dependent), medical consultation, having regular health checkups, and private medical insurance. We observed significant differences.” (Page 5, line 173-176).
“3.3. Status of Non-Participation in BCS” (Page 9, line 217).
“Table 4 shows the frequency of BCS non-participation by attribute.” (Page 9, line 218).
“Table 4. Psychological and personal characteristics affecting non-participation in BCS.” (Page 9, line 236).
“The correlation between BCS and personal characteristics was r<0.7 (|r|<0.7). There were no multicollinearity problems.” (Page 10, line 255-256).
“Excluding medical insurance (dependent), we conducted a logistic analysis with BCS (1: Participation in BCS, 0: Non-participation in BCS) as the dependent variable.” (Page 11, line 262-263).
Point 19: lines 236-7. It is not clear how the authors obtained |r|<0.7 for correlation of BCS with personal characteristics. The interpretation of this estimate is also not clear. The exact value should be reported. r<0.7 could be 0.1 or 0.65. If this is reporting on |r| across all characteristics, then a range of |r| for the variables should be reported.
Response 19: Many thanks for your helpful comments. We have added the text as follows: “Correlation coefficients are presented in Supplementary Material Table S2, and Figure 2 was constructed using the coefficients of the correlation between BCS, personal characteristics and perceptions of harms. The correlation coefficients between BCS and perceptions of harms are ranged from -0.01 (BCS vs. age) to 0.27 (BCS vs having regular health checkups). The coefficients of the correlation between age and personal characteristics ranged from -0.33 (age vs. marital status) to 0.05 (age vs. work status). The coefficients of the correlation between marital status and personal characteristics ranged from -0.36 (marital status vs. medical insurance (independent)) to 0.23 (marital status vs. private medical insurance). The coefficients of the correlation between work status and personal characteristics ranged from -0.14 (work status vs. educational background) to 0.60 (work status vs. medical insurance (dependent)).
The correlation between BCS and personal characteristics was r<0.7 (|r|<0.7). There were no multicollinearity problems.” (Page 10, line 244-256).
Point 20: It states that Figure 2 reports correlation coefficients. But the line above says it is logistic analysis. Also footnote should be included under Figure 2 to explain how to interpret the graph and the meaning and significance of the various colors and circles.
Response 20: Many thanks for your useful comment. We have added the text as follows: “Figure 2. Coefficient of the correlation between BCS consultation and personal attributes. Blue circles represent positive correlations (0.0 to 1.0) between variables and red circles represent negative correlations (-1.0 to 0.0) between variables. The darker the color, the stronger the correlation.” (Page 10, line 258-260).
“Excluding medical insurance (dependent), we conducted a logistic analysis with BCS (1: Participation in BCS, 0: Non-participation in BCS) as the dependent variable.” (Page 11, line 262-263).
Point 21: Decision to get BCS should be a personal decisions based on person risk factors, and a balance of benefits vs. harms. It seems the authors were only emphasizing the harms in this article.
Response 21: Many thanks for your helpful comment. As you pointed out, acceptance is a personal decision. Personal decisions require the full input and understanding of professionals. However, the harms are greater than the benefits of BCS in young people. Next, Tables 2 to 4 of this study show that young adults are characterized by poor awareness of harms and are willing to undergo medical checkups for " my own health" if they have the opportunity to do so. From Table 5 and Table 6, it is clear that the BCS screening behavior is based on the decision of whether or not to undergo BCS screening themselves, along with the improvement of harm awareness. Therefore, we hope that this study, based on the analysis of the current situation, will correctly convey the harms in the younger age group, and that the examinees themselves, with sufficient knowledge, will consider the balance between the benefits and harms of taking BCS checkups and then take action.
Point 22: Extensive editing of English language and style required.
Response 22: Thank you for this remark. A native English professional from our Hiroshima University Writing Center performs extensive English editing. Finally, we have corrected the spelling and reformulated some sentences to improve the language throughout the manuscript.
Reviewer 3 Report
Cui et al. discussed the growing trend of breast cancer screening (BCS) in young people, which is associated with radiation exposure, and the harmful effect may outweigh the benefits of BSC in preventing breast cancer. The authors gathered data from an online survey and demonstrated the disparity in knowledge and awareness across age groups. They also argue that their health insurance provider, employer, and other relevant parties should educate and inform people of all ages—particularly young people—about the advantages and disadvantages of BSC.
Some issues require the author's attention:
1. Data supporting the detrimental impact of BSC, which serves as the foundation for the entire manuscript, must be provided.
2. Table 1 takes up one and a half pages and, aside from column 1 (Data), most of the points in columns 2 (Period) and 3 (Source) are redundant, making it difficult to comprehend its significance. This is covered again on page 4.
Author Response
Cui et al. discussed the growing trend of breast cancer screening (BCS) in young people, which is associated with radiation exposure, and the harmful effect may outweigh the benefits of BSC in preventing breast cancer. The authors gathered data from an online survey and demonstrated the disparity in knowledge and awareness across age groups. They also argue that their health insurance provider, employer, and other relevant parties should educate and inform people of all ages—particularly young people—about the advantages and disadvantages of BSC.
Point 1: Data supporting the detrimental impact of BSC, which serves as the foundation for the entire manuscript, must be provided.
Response 1: We thank the reviewer for this insightful suggestion. We have added and revised the text as follows: “The risk of a false-positive MG is about 20% for women in Europe who undergo biennial screening between 50 and 69 years of age is about 20% over that period, and the risk of undergoing a biopsy as a result of a false-positive screening is 3% [7]. In the United States, the 10-year false positive rate is 30%, with 50% of women experiencing a false positive once [8-9]. On the other hand, in Japan, the rate of false positives is reported to be around 10% per examination [10].”(Page 1, line 37-42).
“Recent studies have shown that the overdiagnosis rate for breast cancer varies widely, ranging from 0% to 54% [12]. Studies based on statistical models consistently estimate the overdiagnosis rates to be below 5% [13-14]. In contrast, observational studies have pub-lished higher estimates ranging from 22% to 54%, depending on the denominator used [12,14-15].”(Page 2, line 46-50). In addition, the following references have also been updated.
[7] Hofvind, S.; Ponti, A.; Patnick, J.; Ascunce, N.; Njor, S.; Broeders, M.; Giordano, L.; Frigerio, A.; Törnberg, S. False-Positive Results in Mammographic Screening for Breast Cancer in Europe: A Literature Review and Survey of Service Screening Programmes. J Med Screen 2012, 19, 57–66, doi:10.1258/jms.2012.012083.
[8] Elmore, J.G.; Barton, M.B.; Moceri, V.M.; Polk, S.; Arena, P.J.; Fletcher, S.W. Ten-Year Risk of False Positive Screening Mammograms and Clinical Breast Examinations. N Engl J Med 1998, 338, 1089–1096, doi:10.1056/NEJM199804163381601.
[9] Hubbard, R.A.; Kerlikowske, K.; Flowers, C.I.; Yankaskas, B.C.; Zhu, W.; Miglioretti, D.L. Cumulative Probability of False-Positive Recall or Biopsy Recommendation After 10 Years of Screening Mammography: A Cohort Study. Ann Intern Med 2011, 155, 481, doi:10.7326/0003-4819-155-8-201110180-00004.
[10] Cancer Information Service. The Disadvantages of Breast Cancer Screening. Available online: https://ganjoho.jp/med_pro/cancer_control/screening/screening_breast.html (accessed on 16 January 2023).
[12] Kalager, M.; Adami, H.-O.; Bretthauer, M.; Tamimi, R.M. Overdiagnosis of Invasive Breast Cancer Due to Mammography Screening: Results From the Norwegian Screening Program. Ann Intern Med 2012, 156, 491, doi:10.7326/0003-4819-156-7-201204030-00005.
[13] de Koning, H.J.; Draisma, G.; Fracheboud, J.; de Bruijn, A. Overdiagnosis and Overtreatment of Breast Cancer: Microsimulation Modelling Estimates Based on Observed Screen and Clinical Data. Breast Cancer Res 2005, 8, 202, doi:10.1186/bcr1369.
[14] Puliti, D.; Duffy, S.W.; Miccinesi, G.; De Koning, H.; Lynge, E.; Zappa, M.; Paci, E. Overdiagnosis in Mammographic Screening for Breast Cancer in Europe: A Literature Review. J Med Screen 2012, 19, 42–56, doi:10.1258/jms.2012.012082.
[15] Zahl, P.-H.; Mæhlen, J.; Welch, H.G. The Natural History of Invasive Breast Cancers Detected by Screening Mammography. Arch Intern Med 2008, 168, 2311, doi:10.1001/archinte.168.21.2311.
Point 2: Table 1 takes up one and a half pages and, aside from column 1 (Data), most of the points in columns 2 (Period) and 3 (Source) are redundant, making it difficult to comprehend its significance. This is covered again on page 4.
Response 2: Thank you for your helpful suggestion. The original Table 1 has been removed with text explaining the source of the data.
We have added and revised the text as follows: “The personal characteristics, BCS participation status, reasons for non-participating in BCS, knowledge of the harms of cancer screening, and HBM used in this study were all derived from the "2018 Attitude Survey on Factors Influencing Women's Cancer Screening Behavior"(Page 3, line 105-108). In addition, other Table numbers have also been updated.
Reviewer 4 Report
Authors performed a nice study on Breast Cancer Screening and Perceptions of Harms in Young Adults. Introduction exposes the context to their objectives and methods are well indicated. Results are clearly described and well discussed with previous literature. Conclusion is also clear for the reader so, in my opinion, this paper is ready to be published.
Author Response
Authors performed a nice study on Breast Cancer Screening and Perceptions of Harms in Young Adults. Introduction exposes the context to their objectives and methods are well indicated. Results are clearly described and well discussed with previous literature. Conclusion is also clear for the reader so, in my opinion, this paper is ready to be published.
Response: We thank the reviewer for the thorough review of our manuscript. We appreciate your encouragement and recommendation for the publication.
Round 2
Reviewer 1 Report
The authors answered the reviewer's questions and made changes to the text of the article. I think that in its present form the article can be recommended for publication.
Author Response
The authors answered the reviewer's questions and made changes to the text of the article. I think that in its present form the article can be recommended for publication.
Response: We thank the reviewer for the thorough review of our manuscript and for your contribution to improve its quality. We appreciate your encouragement and recommendation for the publication.
Reviewer 2 Report
I would like thank the authors for their detailed responses to my questions/comments. I have some additional questions and suggestions regarding their responses. Please the detailed suggestions below.
Point 2: More information about how the online survey is conducted is needed. The method section should describe how participants are selected and administered the surveys, the response rates etc.
Response 2: Thank you very much for your valuable comments. We have added more information in the manuscript as follows: “The online survey title was "2018 Attitude Survey on Factors Influencing Women's Cancer Screening Behavior." The survey used "registered monitors" owned by an online research company (Hiroshima, Japan). Samples were drawn according to the percentage of the population (see 2015 Census) in each age category (20s, 30s, 40s, 50s, and 60s). The survey was stopped when the target sample for each category was reached. A sample of approximately 3,000 persons was deemed adequate.” (Page 3, line 98-103).
Thank you for the response. This description should also include who are the targeted population and inclusion criteria. For example, is it all women in Japan? Of what age? Any requirement for work status? Additionally, it should also include information about how the survey was sent to potential participants. Was it posted through an app, website, etc.? It would help assess the representativeness of the participants in the study. The information above described how age distribution was matched to population distribution, but not other aspects that would make it representative of the population.
Point 4: It’s not clear if the logistic regression results reported in table 6 are for risk of having BCS or not having BCS. This needs to make clear both in the method section and related results and tables.
Response 4: Many thanks for your thoughtful comments. We have added and revised the text as follows: “The flow of logistic analysis in our study was as follows: (i) We conducted correlation analysis based on the results of the χ2 test (p<0.05) and the perception of harms. (ii) We conducted a logistic regression analysis with the BCS (1: Participation in BCS, 0: Non-participation in BCS) as the dependent variable, and the variable with |r|<0.5 obtained from the correlation analysis and the seven factors of HBM as explanatory variables." (Page 4, line 152-157).
Thanks. However, this statement is still not clear to me, “(i) We conducted correlation analysis based on the results of the χ2 test (p<0.05) and the perception of harms.” Do you mean the correlation analysis was the correlation between perception of harms and patient factors? Since factors associated with getting BCS screening and perception of harm may be different, it is not clear why the correlation analysis was not done for both BCS and perception of harms to be used in the model for BCS and model for perception of harm separately.
“Excluding medical insurance (dependent), we conducted a logistic analysis with BCS (1: Participation in BCS, 0: Non-participation in BCS) as the dependent variable." (Page 11, line 262-263).
This sentence is very confusing to me. Please explain why medical insurance (dependent) variable needs to be excluded. Please also see my other comment about creating a table to provide definitions of these covariates.
Point 7: The survey asks whether a participant knows the harms and the attitudes towards knowing the harms of BCS. But it does assess their actual knowledge of harms. It is unknown whether participants have a correct understanding of harms from BCS. And the harm questions are also not BC specific.
Response 7: Thanks for your indepth opinion. As you indicated, there are limits to the specificity of the disadvantage and the ability to make an objective determination. First, the purpose of this study is to understand whether young people are receiving BCS, knowing that the harms of BCS exist. It was also to ascertain the association between the perception of BCS disadvantage and BCS screening behavior. Verifying the specifics of harms are difficult; knowledge assessment without knowing whether or not people are aware of the existence of harms in prior surveys involves the risk that the survey will fail. If no one believes that they know of any disadvantage, no answer will be given. This may not be good for research evolution. Therefore, this study first asked about perceptions of subjective disadvantage. Based on the results of the study to date, we plan to quantify the harms and deepen the study in the future. The ultimate goal of the study was not to know the harms of breast and cervical cancer alone, but to understand the overall harms and to generate screening behavior. We have added and revised the text as follows: “Finally, the harms of BCS, such as decreases in life expectancy and longevity, are subjective responses, and the harms need to be quantified and explored in depth. In addition, the Japanese female labor force is characterized by a bias in employment status by age, so it is necessary to consider how to adjust for this."(Page 14, line 374-378).
Thank you for the authors’ response. Sorry I meant “But it does NOT assess their actual knowledge of harms.” This limitation should be revised, “Finally, the harms of BCS, such as decreases in life expectancy and longevity, are subjective responses, and the harms need to be quantified and explored in depth.” There is little difference between decreases in life expectancy and decrease in longevity. The wording seems redundant. I think any BCS policy goal is to help women acquire correct information about harms and benefits in order to make the best decision for themselves. There are no studies confirming that BCS shortens life expectancy. Having such misunderstanding of the harm from BCS is dangerous and could lead to delayed or missed screening that result in premature death from cancer. Thus, it is important to state as a limitation that this study only asked whether they think they know the harms or interested in knowing the harms, it did not assess their actual knowledge of harms and the correctness of that knowledge.
Point 8: Lines 130 and 135. Cronbach's alpha is a measure of internal consistency, that is, how closely related a set of items are as a group. It is not clear why higher score indicates that participation in screening was perceived more beneficial. Please clarify.
Response 8: Many thanks for your useful comment. We have revised and added the text as follows: “The larger the Cronbach's α coefficient, the more reasonably the variables constructed on the basis of the factors obtained are considered to be represented.” (Page 3, line 143-145).
Thank you for the authors’ response. However, the revised statement is still confusing to me. Please clarify the following, “the more reasonably the variables constructed on the basis of the factors obtained are considered to be represented”.
Point 10: It is not clear why the authors chose |r|<0.5 as a threshold for variables included in the logistic regression. A correlation of |r|>=0.5 is considered large correlation. The variables that have high correlation with receipt of BCS may be different from those associated with presence of interest to know about harm. This should be tested separately for receipt of BCS and harms. In line 150, it is not clear what the p<0.05 mean in that sentence.
Response 10: Thank you for your questions. In general, regression models often suffer from multicollinearity problems. There is now an index VIF (Variance Inflation Factor) that measures multicollinearity. In general, if the VIF statistic is greater than 10, there is a possibility that multicolinearity exists. Therefore, it is necessary to check if the VIF is greater than 10. The VIF can be calculated using the correlation coefficient between explanatory variables with the following formula.
VIF=1/(1-r^2 )
※ris the correlation coefficient.
※For example, r =0.1 (VIF=1.01), r =0.5 (VIF=1.33), r =0.9 (VIF=5.26)
The higher the correlation coefficient, the greater the VIF. In addition, the correlation coefficient is the coefficient of the correlation between r ranging from -1 to 1, where r.0.0~0.2(Hardly any correlation), 0.2~0.5(Somewhat correlated), 0.5~0.7(Quite a correlation), 0.7~1.0(Strong correlation). The study was summarized within 10 VIFs by selecting variables that were somewhat correlated |r|<0.5 among the variables.
Thank you for the detailed explanation. However, VIF measures the multi-collinearity in a regression model. The R2 is not the correlation coefficient from bivariate correlation of BCS with each personal factor variable. It should be the R2 from the regression with all variables.
To clarify the association between breast cancer screening and disadvantage, this study introduced the BCS influence factor as an explanatory variable for harms.
Page 5, line158 p<0.05 is a significant result obtained from logistic regression.
Thank you. Does disadvantage here mean the harms from BCS? It would be helpful to use consistent terminology to avoid unnecessary confusion.
Point 12: The health belief questions were not specific to BCS, but for women cancer screening in general. The attitudes may be different for screening of BC vs. cervical cancer.
Response 12: As you pointed out, a detailed investigation would be different. However, in this study, we first considered it largely as cancer specific to women as an exploratory study for hypothesis generation.
Thank you for authors’ response to my question. However, it did not fully address my concern. This should be acknowledged as a limitation of the study. That is, the health belief questions were not specific BCS but was interpreted as if they are about BCS.
Point 13: Lines 164-167. The difference between medical insurance and private insurance is unclear. It seems private insurance and medical insurance overlaps and one should be included but not two.
Response 13: Thank you for your questions. Medical insurance and private insurance in this study cannot be merged. This section describes the health situation in Japan. The reasons for this are: ①Medical insurance is Japan's public medical insurance. Public health insurance is compulsory for all citizens. In other words, it is a system in which citizens support each other so that everyone can receive medical care with peace of mind. Private insurance, on the other hand, provides hospitalization and surgical benefits as the main policy, although the coverage varies by the insurance company. The major distinction between the two insurance policies is shown in Table 1. ②The questions for medical insurance and private insurance are different and cannot be merged. It is preferable to describe it as a characteristic of the health care system in the text. However, it would be redundant and should be explained only in the peer review reply.
Table 1. Distinction between medical insurance and private insurance
|
Medical Insurance |
Private Insurance |
|
|
Eligibility |
Mandatory Enrollment |
Voluntary Enrollment |
|
Purpose of Membership |
Stability of people's lives |
Co-pay coverage |
|
Insurance Premium |
Depends on income |
Varies by age, gender, and coverage |
|
Benefit |
Benefit in Kind |
Receipt of insurance proceeds |
|
Types and Systems |
Health insurance, high-cost medical care, etc. |
Medical insurance, cancer insurance, etc. |
Thank you for the detailed explanation. It is very helpful. Since the readers are international and likely are not familiar with Japan’s healthcare system, it would be helpful to explain this in the text also. Given the definition of some other variables (e.g. medical insurance (dependent), medical consultation) are also not clear to the readers, I would suggest to have supplemental table explain each of the covariates included.
Point 14: Table 3. Some are reporting row %. Should report column % for all variables.
Response 14: Thank you for your helpful suggestions. Corrected % in Table 2 (original Table 3 ). (Page 6, Table 2. Associations between BCS behaviors and personal characteristics/harms).
We have added the text as follows: "Observing the number of respondents by item showed that the numbers for an awareness of harms were 12 (24.5%) and 37 (75.5%) for those who did or did not receive BCS, respectively. On the other hand, 194 (24.2%) and 607 (75.8%) of the examined and unexamined respondents, respectively, were concerned about the harms of BCS.” (Page 5, line 177-180).
Thank you for making the changes. Is there a difference between receive BCS and examined? Why 12 is 24.5% for those who did receive BCS, but 194 was 24.2% for those who are examined? These suggest that the %’s are out of different base populations. But receive BCS and examined, based on the description in the text, should be the same. Please clarify. Same question for the %’s of the not received BCS and not examined.
Point 19: lines 236-7. It is not clear how the authors obtained |r|<0.7 for correlation of BCS with personal characteristics. The interpretation of this estimate is also not clear. The exact value should be reported. r<0.7 could be 0.1 or 0.65. If this is reporting on |r| across all characteristics, then a range of |r| for the variables should be reported.
Response 19: Many thanks for your helpful comments. We have added the text as follows: “Correlation coefficients are presented in Supplementary Material Table S2, and Figure 2 was constructed using the coefficients of the correlation between BCS, personal characteristics and perceptions of harms. The correlation coefficients between BCS and perceptions of harms are ranged from -0.01 (BCS vs. age) to 0.27 (BCS vs having regular health checkups). The coefficients of the correlation between age and personal characteristics ranged from -0.33 (age vs. marital status) to 0.05 (age vs. work status). The coefficients of the correlation between marital status and personal characteristics ranged from -0.36 (marital status vs. medical insurance (independent)) to 0.23 (marital status vs. private medical insurance). The coefficients of the correlation between work status and personal characteristics ranged from -0.14 (work status vs. educational background) to 0.60 (work status vs. medical insurance (dependent)).
Thank you. Please correct this statement, “The correlation coefficients between BCS and perceptions of harms are ranged from -0.01 (BCS vs. age) to 0.27 (BCS vs having regular health checkups).” These seem to be the correlation between BCS and personal characteristics. Also, does Figure 2 examine correlation of BCS with perception of harms?
The correlation between BCS and personal characteristics was r<0.7 (|r|<0.7). There were no multicollinearity problems.” (Page 10, line 244-256).
It would be helpful to report VIF to support this statement.
Point 20: It states that Figure 2 reports correlation coefficients. But the line above says it is logistic analysis. Also footnote should be included under Figure 2 to explain how to interpret the graph and the meaning and significance of the various colors and circles.
Response 20: Many thanks for your useful comment. We have added the text as follows: “Figure 2. Coefficient of the correlation between BCS consultation and personal attributes. Blue circles represent positive correlations (0.0 to 1.0) between variables and red circles represent negative correlations (-1.0 to 0.0) between variables. The darker the color, the stronger the correlation.” (Page 10, line 258-260).
“Excluding medical insurance (dependent), we conducted a logistic analysis with BCS (1: Participation in BCS, 0: Non-participation in BCS) as the dependent variable.” (Page 11, line 262-263).
Thank you for adding the footnote. What does BCS consultation mean? Does that mean the woman did not get the actual screening but only visited the doctor for consultation? Please clarify.
